# Manufacturing of Soft Contact Lenses Using Reusable and Reliable Cyclic Olefin Copolymer Moulds

**DOI:** 10.3390/polym14214681

**Published:** 2022-11-02

**Authors:** Christopher Musgrave, Lorcan O’Toole, Tianyu Mao, Qing Li, Min Lai, Fengzhou Fang

**Affiliations:** 1Centre of MicroNano Manufacturing Technology (MNMT-Dublin), University College Dublin 4, D14 YH57 Dublin, Ireland; 2State Key Laboratory of Precision Measuring Technology and Instruments, Laboratory of MicroNano Manufacturing Technology (MNMT), Tianjin University, Tianjin 300072, China

**Keywords:** contact lenses, manufacturing, cyclic olefin copolymers, surface characterisation

## Abstract

We present experimental evidence of reusable, reliable cyclic olefin copolymer (COC) moulds in soft contact lens manufacturing. The moulds showed high performance surface roughness characteristics despite >20 kW exposure to 365 nm ultraviolet (UV) light from repeated use. Ultra-precision manufacturing techniques were used to fabricate transparent COC mould inserts and to produce soft contact lenses from liquid monomer compositions. Both polymer and silicone hydrogels were fabricated with more than 60 individual uses of the moulds. White light interferometry measured the surface roughness (Sa) of the COC moulds to be almost unchanged before and after repeated use (Sa 16.3 nm before vs. 16.6 nm after). The surface roughness of the prototyped lenses and that of commercially available soft contact lenses were then compared by white light interferometry. The surface roughness of the lenses was also nearly unchanged, despite undergoing more than 60 uses of the COC moulds (lens Sa 24.4 nm before vs. after Sa 26.5 nm). By comparison the roughness of the commercial lenses ranged from 9.3–28.5 nm, including conventional and silicone lenses, indicating that the reusable COC moulds produced competitive surface properties. In summary, COC moulds have potential as reusable and reliable mould inserts in the manufacturing of soft contact lenses, yet maintain high quality optical surfaces even after sustained exposure to UV light.

## 1. Introduction

Manufacturing technology has been crucial at determining the rate and ceiling of human technological advancement. This is exemplified by human journeys through various stages of manufacturing industry revolutions (Manufacturing I and II, and soon to be Manufacturing III) [1,2,3]. The impacts of the revolutions are seen every day across many sectors from medical devices, to semi-conductor production. This includes contact lens manufacturing, which utilises many of these technological advancements [4]. Recently, sustainability is a hot topic, which applies to manufacturing technology- smarter, more efficient methods of production [5,6,7]. Sustainable manufacturing can be considered less wasting of energy and materials, involving less use of polluting materials, and more use of materials from reliable sources. Some particular concerns are the excessive use of micro-plastics and plastic packaging, which have a significantly detrimental environmental impact [8,9,10,11]. Plastics often have long degradation times and are taken up by the food chain. This directly relates to the contact lens (CL) industry, as contact lenses are small plastics, with individual plastic packaging [12,13]. With over 150 million users of CLs worldwide, there is a looming question of sustainability for the industry.

Soft contact lenses are medical devices used for corrective vision in place of eye glasses. They are composed of hydrophilic polymers, hydrogels, that absorb water to produce a transparent, malleable material that can be placed on the eye [14,15]. Modern interests and applications of CLs include myopia control, but also as an efficient drug delivery mechanism [16,17,18,19,20]. Consequently, there is a large interest in the development and manufacturing of new, more functional CLs, including those having bactericidal properties [21]. To achieve these goals, both material development and manufacturing technologies are relevant. There are several notable material developments, from the first soft poly-2-hydroxyethyl methacrylate (polyHEMA) lenses to modern silicone hydrogels. For example, the increase of oxygen permeability of lenses has been one of the most standout developments from early lenses to today’s lenses. Well-documented manufacturing innovations can be highlighted by the Lightstream™ process to manufacture Nelfilcon lenses, or plasma treatment to improve the hydrophilic properties of silicone hydrogel lenses [22,23]. These improvements have seen the CL industry grow, with the development of new materials and more efficient manufacturing methods. Given the startling projections on the state of global eye health by 2050, and, specifically, issues such as high myopia and associated diseases, even more developments are necessary to meet future challenges. Hence, strides in materials technology, as well as an increase in the scale of manufacturing, as more people use CLs as a treatment option, are called for. However, there will also be an associated use of plastics, adding to the environmental problem. Therefore, there is a need for new techniques and materials that can improve the sustainability of contact lenses and their manufacturing to meet the challenges of eye health. 

CLs are typically manufactured in a high precision injection moulding process. The process involves the use of polymers, such a polyethylene (PE) or polypropylene (PP), for moulding and packaging of the lens [14,24]. These polymers are typically only used once. The mould insert is shaped to a chosen lens power, base curve etc., inside a larger mould, then removed for post-processing and packaging. Lenses are fabricated using a fast ultraviolet (UV) light curing process to solidify the lens formulation, allowing for high manufacturing throughput. For lens moulding, a single use mould insert ensures a level of repeatability of lens manufacturing and each can be customised to a particular power or curvature. Furthermore, a single use insert does not run the risk of UV light degradation through repeated exposure. Once the lens is moulded, it undergoes processes of cleaning, swelling and sterilization that are essential for safe use. Whilst highly efficient and cost effective, the industry is arguably not currently fulfilling sustainability needs. This issue has only recently entered discussion within the research and industrial community [12,25,26,27]. Many of the big manufacturers have started recycling initiatives for the packaging and lenses [27]. However, given that CLs are typically crosslinked polymers they are generally very difficult to truly reprocess, unless new approaches to lens materials are adopted [28,29]. Therefore, one of the biggest challenges is what to do with the contact lens and the packaging after use; with some new materials being considered. For manufacturing, new methodologies and materials could be introduced that maintain high throughput of manufacturing but improve sustainability. One solution could be a reusable mould insert, instead of a single use material. A key requirement for a reusable mould would be to maintain quality optical surfaces after significant UV light exposure i.e., multiple uses. Secondly, the material would need to be reprocessed to allow different lens shapes and powers to be produced. Finally, the material must not negatively impact the biocompatibility of the lenses i.e., leeching of volatiles into the lens etc. One such material that fits these criteria is cyclic olefin copolymer (COC). It is a material already gaining significant use in biomedical applications but has yet to gain visible traction in the CL industry.

COC is an ethylene-norbornene derived polymer with superior properties than that of similar transparent plastics; including higher modulus, excellent optical properties and moisture resistance [30]. COC has glass-like properties which lend themselves for use as lenses in optical-grade applications [31]. The polymer is easy to injection mould, which means it can realise many different applications, such as lab-on-a-chip, whereby micrometre features can be accurately replicated [32]. COC can also be reprocessed; for example, TOPAS^®^ is a branded version of COC that has recycling certification [33]. The polymer can also be further modified for additional/improved functionality, including increased hydrophilic properties [34]. COC can also be sterilized, which has seen its use grow in medical applications [35]. This sterilization property meant COC-based devices have been used in testing procedures for COVID-19, or as an effective replacement for borosilicate glass syringes. However, there are no reports of using this useful material in the context of contact lens manufacturing. COC could be used as a mould insert as a like for like replacement for a PE/PP mould insert, or to replace optical elements that are usually composed of quartz. The extent to which COC can be modified offers the possibility of successful integration into CL moulding methods, with the aim of improving manufacturing sustainability.

In this report a contact lens mould was fabricated from COC, then used to produce soft contact lenses. The surface roughness (Sa) of the both the COC mould and soft contact lenses were analysed before and after repeated use. White light interferometry measurements showed that the mould was nearly unchanged after more than 20 kW 365 nm of UV light exposure, with little negative impact on the surface roughness of the mould or the lenses. For comparison, the surface roughness of commercial lenses was measured. The evidence suggested that COC could be used in contact lens manufacturing processes as the reusable mould part.

## 2. Materials and Methods

### 2.1. Materials and Mould Manufacturing

Inhibitor removal beads eliminated inhibitors, hydroquinone and monomethyl ether hydroquinone, from the monomers prior to use. A typical hydrogel lens contained a mixture of 2-Hydroxyethyl methacrylate (HEMA) (Sigma-Aldrich, St. Louis, MO, USA), Dimethyl acrylamide (DMA) (Sigma-Aldrich) and ethylene glycol dimethyacrylate (EGDMA) (Sigma-Aldrich. The silicone hydrogel contained both HEMA and DMA, and also n-vinyl pyrrolidone (NVP) (Sigma-Aldrich) and 3-[Tris(trimethylsiloxy)silyl]propyl methacrylate (TRIS) (TCI Chemicals). The photoinitiator was 2-Hydroxy-4′-(2-hydroxyethoxy)-2-methylpropiophenone (Irgacure 2959) (Sigma-Aldrich) (0.5 wt%). The COC (480R, ZEON, Japan) moulds were manufactured from 40 mm diameter by 14.7 mm thick blocks (male mould) and 40 mm diameter by 11.25 mm blocks (female mould). The COC was used as received.

A typical polymer hydrogel formulation was composed of 72.6:27:0.4 wt% HEMA:DMA:EGDMA. A typical silicone hydrogel formulation was composed of 29.5:20:35:15:0.5 wt% HEMA:DMA:NVP:TRIS:EGDMA. These formulations represent major classes of soft contact lenes used commercially. Therefore, the aim was to study the impact of these formulations during manufacturing with COC moulds.

Commercially available contact lenses, comprised of fanfilcon A, stenfilcon A, omafilcon A, nelfilcon A and etafilcon A, were measured using a white light interferometer. Lenses were removed from the sealed packaging and placed in deionized water (DI water) for 24 h to remove any species that could influence the interferometry measurements e.g., crystalizing salts during lens dehydration.

The COC mould has an aspheric lens profile, two exhaust shafts and screw sockets (Figure 1A). It was produced by a single point diamond turning (SPDT) technique, which is widely used to machine aspheric and freeform optical surfaces with nanoscale surface roughness and submicron-scale form accuracy. As shown in Figure 1B, the mould workpiece was mounted on an air bearing spindle and spun during the processing. With precisely relative motion of the diamond tool in the feed and infeed directions, the surface materials were accurately removed and the removal scale, e.g., undeformed chip thickness, could be as small as several nanometers, which was beneficial for the nanoscale surface roughness. The complex form of the machined surface could be realised with the well-designed relative motion model of diamond tool. A clampable steel mould was designed to house the COC mould and UV light, as shown in Figure 1C.

### 2.2. Lens Irradiation Conditions and Lens Fabrication

A 365 nm ultraviolet (UV) light (UVET NSC-4) was used to form all lenses with the COC mould. The working distance was 39 mm from the UV light lens to the middle of the mould, which corresponded to 0.73 W/cm^2^ light intensity, according to the technical description of the UV light and inverse-square law. The spot size was 13.5 mm. Each lens was cured in about 330 s.

The COC mould inserts were placed into the corresponding clampable steel mould. Then, a polymer or silicone lens formulation was added to the female mould, which was then closed and secured by the clamps (Figure 1C). The system was then exposed to UV light until solidified xerogel formed. DI water was then inserted into the exhaust shafts of the COC mould to induce lens swelling and release from the mould surface which took between 60 min to several days, depending on the formulation. The lenses were finally rinsed in IPA/H_2_O before final swelling in DI water.

### 2.3. White Light Interferometry

A White Light Interferometer (WLI) (Bruker NPFlex) was used to measure the surface roughness of the COC moulds and lenses. The WLI was performed in Vertical Scanning Interferometry (VSI) mode, all images were then processed in Vision 64 with either a Terms removal-F operator and Stylus analysis (7 lengths) filer, or Gaussian regression filter and S (roughness) height parameters calculation. The filters removed any curvature of the moulds and lenses to obtain a flat surface for roughness calculation. The contact lenses required care and attention when measuring. Lenses were cut into smaller pieces using a clean scalpel to mitigate curvature. To mitigate the impact of surface debris on the measurements, the lenses were cleaned in an IPA/DI water mix under ultrasonication, then placed back into DI water overnight before measurement. Excess water was removed and the lens placed on a clean glass slide. Measurements were performed several times. If excess water remained on the surface, the roughness value was unrealistic (below that of the mould). Conversely, if the lens dehydrated excessively the surface roughness value was much greater than the mould. Dehydration was clear when the lens was visibly curling off the surface of the slide. Typically, lens measurements were strictly required to be completed within several minutes of removal from the storage vial.

## 3. Results

Figure 2 demonstrates a typical soft contact lens made using the COC moulds after cleaning and swelling. The prototyped lenses typically retained their shape on manual handling, as demonstrated by placement on a finger, which indicated reasonable mechanical properties.

Figure 3 and Figure 4 show white light interferometry images of new and used COC moulds. The male (Figure 3) and female (Figure 4) moulds were measured individually. due to the difference in UV light pathlength, with the female mould receiving less UV light than the male mould. The surface roughness of the male COC mould remained about the same at 18.6 nm before (Figure 3A) and 18.6 nm after (Figure 3B) sustained UV exposure. Figure 3b shows some surface damage, likely scratches from user handling/cleaning. The surface roughness of a female mould was also the same at 16.3 nm before (Figure 4A) and after (Figure 4B) repeated manufacturing of soft contact lenses. The COC mould surface roughness data is summarised in Table 1.

Figure 5 shows white light interferometry images of two different soft contact lens surfaces from a new COC mould (Figure 5A) and after significant use of the mould (5B). The roughness of the lenses was measured at 24.4 nm from a new mould and 26.5 nm for a lens from a used mould (Table 2). The surface roughness of the prototyped contact lenses was compared to commercially available lenses, ranging from silicone to conventional hydrogel materials (Table 2).

## 4. Discussion

Prototyping of soft contact lenses was successful using COC moulds (Figure 2). Often, research papers regarding new lens materials do not show the manufactured lens for scale or optical quality. The COC moulds were smooth enough to produce optical quality finish on the lenses. The lens Sa values, according to WLI, were also virtually unchanged from an unused mould (24.4 nm) and after significant use (26.5 nm). Therefore, the surface roughness was more related to the mould surface and not the chemical composition. The WLI data showed that the roughness of the lenses was higher than the mould. There were several possible explanations for this. One explanation was that the lenses dehydrated during preparation for interferometry measurements. This was almost unavoidable, as the interferometer required the top surface to be the surface of the lens, which ruled out using any hydration devices to keep the lens wet. It could be assumed that the roughness of a non-dehydrated lens was the same as the mould. Another plausible explanation was that the accumulation of debris increased the roughness overall. Whilst care was taken to clean samples, the interferometry images appear to show some features that look like debris and could account for the error as large debris were included in the measurement. Moreover, these WLI measurements were further evidence for its use in CL measurements, which is relatively uncommon [36]. However, the roughness results were essentially the same for the purposes of mould repeatability and reliability. The roughness values for the lenses moulded by COC moulds were then compared with the roughness of commercial lenses (Table 2). The commercial lens roughness varied by material type, conventional or silicone hydrogel, but, overall, was not dissimilar to the lenses fabricated with COC moulds. The lowest roughness was for nelfilcon A, a conventional hydrogel, at 9.3 nm in Sa, and the highest for omafilcon A, a silicone hydrogel, at 28.5 nm in Sa. We could possibly infer from our experiments that the industrial PP/PE moulds were very smooth; close to, if not better than, the roughness of the lens itself (Table 2). Ideally, in the future we could clarify this with an industrial partner. Thus, the reusable COC moulds produced competitive surface roughness compared with other single-use lens mould manufacturing processes. In terms of clinical relevance, the values of the lens surface roughness likely matter less than other aspects, such as tear film breakup time and oxygen permeability. The impact of a wider tolerance range on roughness is that manufacturers have more flexibility in mould manufacturing, whereby more precision is usually more expensive.

An additional interesting aspect of the lens moulding was the lens release time. The exhaust shaft was filled with DI water to induce lens swelling which released them from the mould without damage. The release time increased if more silicon-containing monomers were used in the lens formulation. In some cases, the release time was several days. There seems to be no reports concerning this for contact lenses but it is important for manufacturing chains. For polymer lenses, the release time was as short as one hour. One likely reason was chemical interactions between the solidified xerogel and the COC. As more Si-containing monomers were introduced the number of hydrophobic groups also increased. These groups had a preferential interaction with COC, and so the lens took longer to release from the mould.

White light interferometry was used to measure the female mould roughness, which was found to be less than 20 nm Sa before and after >20 kW UV light exposure to the mould. The COC moulds (male and female) were resistant to multiple cycles of cleaning by solvents (ethanol and IPA) and contact by a cotton swab. In particular, the roughness of the female mould was unchanged, whereas there was some suggestion of a change on the male mould, based on the increased error of the measurement. This could be explained by inclusion of scratches/debris on the measured areas, which increased the Sa value and is visible in Figure 3B. It was probable that a harder material was accidently rubbed on the surface during the many uses/cleaning cycles of the moulds. It is unlikely this problem would occur in a strict manufacturing environment where the engineering controls are much tighter. Strictly speaking, 20 kW exposure would correspond to the male mould light exposure, and would be somewhat lower for the female mould, given the greater distance from the light source. However, the stylus analysis showed that there was no negative impact by the UV light on the male mould surface despite the closer distance. With this evidence there should be a discussion around how COC could be used in a scaled-up manufacturing process. Instead of using PE/PP as inserts, then doubling these up as packaging, COC moulds could be used for moulding, and then another sustainable source used for packaging of the lenses. Some materials of interest here could be biodegradable plastic/cardboard, provided they can meet the standards required (sterilization, shelf-life etc). COC is a good material for injection moulding so should easily form the plus or minus powered lenses, base curves etc., in the same way as current moulding materials. COC can be repurposed because COC is not a cross-linked polymer, therefore mould inserts could be reprocessed if different lens powers or shapes are required, offering the potential for reduced environmental impact as COC is reused. In practical terms, this would mean a large number of COC moulds would be used to keep pace with high volume manufacturing.

One question remains as to what the final lifetime of a COC mould would be? More systematic tests would be needed to answer this question than were possible here. Given the scale of lens production, it would be essential to know the lifetime for a COC mould. The data here suggested the minimum number of uses could in the order of hundreds, based on the 6 h total exposure time of the mould. Although this might not be enough for high-volume manufacturing, the use of a mould several hundreds of times would already significantly decrease any single use plastics. This is without any further modification of the COC which could extend the lifetime of the mould. There are some studies on the effect of constant UV light on COC thin films (25 µm), which suggest it would be negatively impacted after 5–10 days [37]. In that study, ASTM D 5208-01, the UV light was set at 340 nm and at an intensity of 0.78 W/m^2^. These effects included a reduction of the maximum strain and modulus. Another saw significant yellowing of 1 mm thick unmodified COC after 1120 h 0.7 W/m^2^ light exposure [38]. Clearly, this would be an issue for a mould over an extended time, but this is without considering UV stabilisers. Moreover, the COC moulds used here were several millimetres thick in the optical zone and were not placed under any stress/strain, other than the clamping force from the mould. As such, it is expected that thicker COC would have a longer functional lifetime than a thin film or 1 mm thick samples. Therefore, the upper end of use for a >1 mm thick mould could be for more than 11,000 lenses produced per mould before significant yellowing of the COC (based on 6 h of use for 60 lenses). Further decreases to curing time could again increase this projected number. The foremost interest of this study was the surface properties, which ultimately have the biggest impact on contact lens moulding. Therefore, it is expected that, whilst there might be UV damage over time to the COC, the crucial detail is the impact of UV light on the surface over extended use. The ultimate purpose is to highlight to researchers interested in contact lens sustainability that there are possibilities for change from a manufacturing perspective.

## 5. Conclusions

In summary, COC mould inserts were used to fabricate soft contact lenses. The surface roughness of the lenses and moulds were measured before and after significant 365 nm UV light exposure. The COC moulds were used to produce soft contact lenses with comparative surface roughness values to commercial lenses, indicating the potential use of the moulds in manufacturing. The surface roughness showed little to no change despite high energy of exposure and repeated use. Both polymer and silicone hydrogel lenses were manufactured using COC moulds with equal success. The COC moulds produced reliable and repeatable surface roughness values that could translate well into higher volume manufacturing of contact lenses, compared to single use polyethylene/polypropylene moulds.

## Figures and Tables

**Figure 1 polymers-14-04681-f001:**
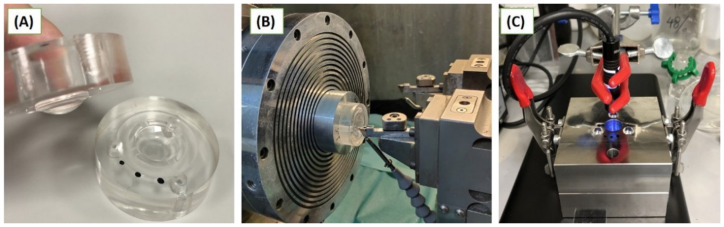
Aspheric COC mould (**A**), SPDT of optical mould (**B**) and moulding system including COC mould and UV light source (**C**).

**Figure 2 polymers-14-04681-f002:**
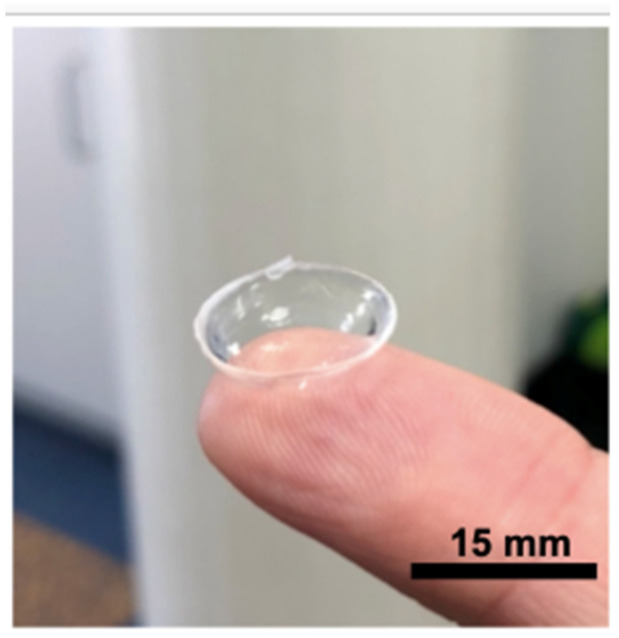
A prototyped soft contact lens from COC moulds after cleaning and swelling on a finger for scale.

**Figure 3 polymers-14-04681-f003:**
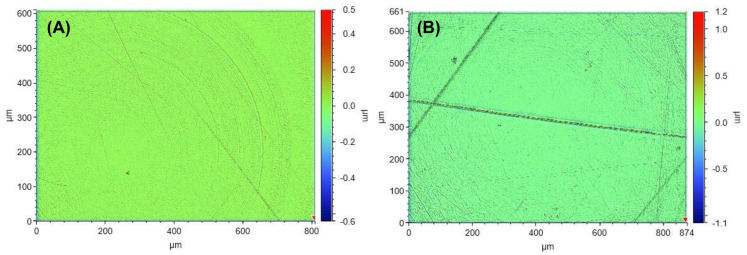
White light interferometer images from a male mould before (**A**) and after significant UV exposure and repeated use (**B**). The coloured legend indicates the height of the roughness in µm.

**Figure 4 polymers-14-04681-f004:**
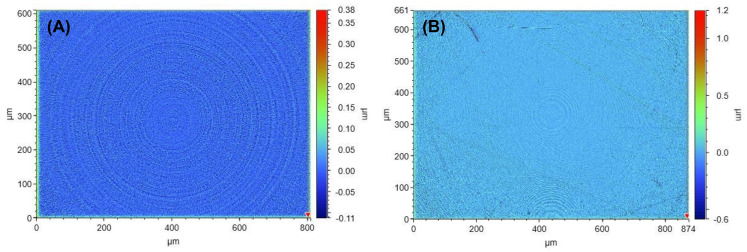
White light interferometer images from a female mould before (**A**) and after significant UV exposure (**B**). The coloured legend indicates the height of the roughness in µm.

**Figure 5 polymers-14-04681-f005:**
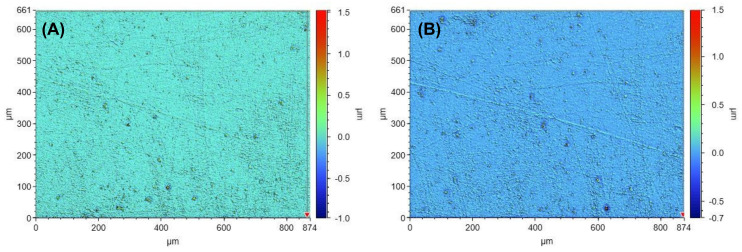
A soft contact lens produced from an unused COC mould (**A**) and a lens produced after significant mould use (**B**). The coloured legend indicates the height of the roughness in µm.

**Table 1 polymers-14-04681-t001:** White light interferometry surface roughness values of COC moulds before and after UV light exposure and two different soft contact lenses.

	Roughness (S_a_) (nm)
Male mould (unused)	18.6 ± 0.7
Female mould (unused)	16.3 ± 1.2
Male mould (>60 uses)	18.5 ± 4.9
Female mould (>60 uses)	16.3 ± 2.3
Lens (unused mould)	24.4 ± 10.6
Lens (>60 uses)	26.5 ± 8.6

**Table 2 polymers-14-04681-t002:** White light interferometry surface roughness of COC moulded lenses and commercial lenses.

Lens Material	Roughness (S_a_) (nm)
COC moulded lens unused	24.4 ± 10.6
COC moulded lens >60 uses	26.5 ± 8.6
Fanfilcon A (silicone hydrogel)	19.6 ± 1.9
Stenfilcon A (silicone hydrogel)	14.2 ± 2.5
Omafilcon A (conventional hydrogel)	28.5 ± 0.7
Nelfilcon A (conventional hydrogel)	9.3 ± 2.8
Etafilcon A (conventional hydrogel)	15.1 ± 1.7

## Data Availability

Not applicable.

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
