# Peer review of "Manufacturing of Soft Contact Lenses Using Reusable and Reliable Cyclic Olefin Copolymer Moulds"

_polymers, 2022, doi:10.3390/polym14214681_

Round 1
Reviewer 1 Report
In the article “Manufacturing of soft contact lenses using reusable and reliable cyclic olefin copolymer moulds” by Christopher Musgrave et al. are investigating the reusability of a robust cyclic olefin copolymer (COC) mould in the manufacture of soft contact lenses.
Notes on the article:
- in the abstract it is not necessary to indicate the words "goal, methods, results, conclusion"
- remake links in the text. Correctly so: [1].
- in the results, in addition to pictures and tables, add a description
- add conclusions
Reviewer 2 Report
This manuscript is proposing the manufacturing of contact lenses moulds made of COC as advantageous to traditional ones made of other polymers. There are a series of points that need to be clarified.
Abstract:
It needs to be rewritten in a more compact and fluid form, rather than as sections.
Also repetitions need to be avoided therein (e.g. the exposure conditions appear four times).
UV needs to be spelled out here too.
Line 54: The abbreviation polyHEMA needs to be spelled out.
Line 100: braded --> branded
Section 2.1.: The COC used here and the way the moulds were manufactured is missing completely.
Line 146: 39mm --> 39 mm
Figure 2: The quality needs to be improved. The lens is hardly visible at all?
Section 3 (Results): This section clearly needs description of the presented figures and tables. For example Figure 3A, 4 and 5 are not referenced in the text at all. Or, here as well as in the Discussion section, the two types of experimental lenses mentioned in the Materials, namely silicone-based and polymer-based are not distinguished or refered to at all.
Discussion: How do the results (i.e. sourface roughness) of the COC moulds compare to traditional moulds made of PP or PE?
Line 277: >1mm --> >1 mm
Round 2
Reviewer 2 Report
The authors have addressed the open points quite well. Indeed Figure 2 as appears in the word version makes sense. In the updated pdf still the old version is present, this needs to be changed.
I am still setting the status to "major revision needed" because of the following reason: The materials of the contact lenses used for the study are described in great detailed in section 2.1, as well as the manufacuring of the moulds. However there is no reference at all to the COC materials actually used for the moulds themselves? One can assume from reference 33 that they were sourced from TOPAS, but even then, TOPAS is offering for example almost 10 different grades for injection moulding? https://topas.com/tech-center/datasheets?field_category_value%5B%5D=injection-molding
The proposal of using COC along with all their suggested advangates is in the core of this paper, so it makes sense to be a bit more specific here?
